# A Randomized Controlled Crossover Lifestyle Intervention to Improve Metabolic and Mental Health in Female Healthcare Night-Shift Workers

**DOI:** 10.3390/nu17213342

**Published:** 2025-10-24

**Authors:** Laura A. Robinson, Sarah Lennon, Alexandrea R. Pegel, Kelly P. Strickland, Christine A. Feeley, Sarah O. Watts, William J. Van Der Pol, Michael D. Roberts, Michael W. Greene, Andrew D. Frugé

**Affiliations:** 1Department of Nutritional Science, Auburn University, Auburn, AL 36849, USA; 2College of Nursing, Auburn University, Auburn, AL 36849, USA; 3School of Nursing, East Tennessee State University, Johnson City, TN 37614, USA; 4Department of Biomedical Informatics, The University of Alabama at Birmingham, Birmingham, AL 35233, USA; 5School of Kinesiology, Auburn University, Auburn, AL 36849, USA

**Keywords:** dietary intervention, night-shift workers, visceral fat, quality of life, circadian rhythm

## Abstract

**Background:** Circadian rhythm disruption caused by shift work alters metabolic and hormonal pathways, which accelerates chronic disease onset, leading to decreased quality and quantity of life. This study aimed to determine whether a practical lifestyle intervention emphasizing nutrition timing and recovery habits could mitigate the metabolic and psychological effects of night-shift work. We conducted a randomized, open-label, crossover trial with two 8-week periods. **Methods:** Female healthcare workers (n = 13) aged 18–50 years with a body mass index (BMI) between 27 and 40 kg/m^2^ and working predominantly night shifts (≥30 h/week for ≥6 months) were randomized. During the 8-week intervention phase, participants received daily text messages with guidance on food, sleep/rest, and physical activity and were provided with whey protein isolate powder and grain-based snack bars to consume during work shifts. The program targeted improved nutrient timing, adequate protein intake, and structured rest without formal exercise training, allowing evaluation of dietary and behavioral effects feasible for this population. Total caloric (~30 kcal/kg lean mass) and protein (2 g/kg lean mass) needs were measured, along with sleep/rest goals of 6–8 h/24 h. Primary outcome measures were change in visceral fat percentage (VF%) by DXA and mental/physical quality of life (RAND SF-12). Secondary outcomes included fasting triglycerides, ALT, blood glucose, LDL, actigraphy, and fecal microbiome. Mixed-design two-way ANOVA was conducted to assess the effects of group (immediate [IG] and delayed [DG]), time (baseline, 8-week crossover, and week 16), and Group × Time (GxT) interactions, and Bonferroni correction was applied to post hoc *t*-tests. **Results:** Eleven participants completed the study. Both groups increased dietary protein intake (*p* < 0.001), and a GxT interaction for VF% (*p* = 0.039) indicated DG reduced VF% to a greater extent (−0.335 ± 0.114% (*p* = 0.003) vs. 0.279 ± 0.543% (*p* = 0.158)). Mental and physical QOL, objectively measured physical activity and sleep, serum lipids and inflammatory markers, and fecal microbiota remained unchanged (*p* > 0.05 for all GxT). **Conclusions:** The findings suggest that targeted nutrition and recovery strategies can modestly improve dietary intake and visceral fat; however, consistent with prior work, interventions without structured exercise may be insufficient to reverse broader metabolic effects of circadian disruption. This trial was registered at ClinicalTrials.gov, identifier: NCT06158204, first registered: 28 November 2023.

## 1. Introduction

Night-shift work with unconventional working hours poses many challenges to maintaining healthy lifestyle habits. The stress and disruption of their circadian rhythm make these workers highly susceptible to poor metabolic health [1,2]. Recently, research has established that normal dietary recommendations alone are not enough to manage their overall health, even when consuming similar caloric amounts to daytime workers [1]. Unfortunately, these shift workers are more likely to become overweight or obese, and they tend to have higher levels of adiposity [2]. In longitudinal studies, obesity and adiposity are associated with the onset of chronic conditions and metabolic complications [3].

It has been proposed that circadian misalignment, along with poor dietary choices, is to blame for these consequences [4]. The circadian rhythm, operating on a 24 h cycle, has the potential to be altered by exposure to light or nutrient intake. This rhythm is ultimately controlled by the suprachiasmatic nucleus (SCN) in the hypothalamus. The SCN is responsible for coordinating the peripheral clocks [5]. Among these peripheral clocks, the liver plays a critical role in maintaining energy balance. Core clock genes, such as CLOCK, BMAL1, PER, and CRY, regulate rhythmic transcription of enzymes involved in gluconeogenesis, lipid synthesis, and bile acid metabolism, aligning hepatic metabolism with feeding–fasting cycles [5,6,7]. When this synchrony between the central and hepatic clocks is disrupted, as occurs during night-shift work, metabolic gene expression becomes desynchronized, contributing to insulin resistance, dyslipidemia, and increased visceral adiposity [6,7]. Experimental evidence indicates that loss of BMAL1 or CLOCK function impairs glucose tolerance, alters NAD^+^/SIRT1 signaling, and disturbs lipid homeostasis, highlighting the molecular link between circadian clock disruption and metabolic disease [6]. The liver is therefore a key mediator through which circadian disruption translates into metabolic abnormalities [5].

With shift work, sleep is often one of the sacrifices many night-shift employees make. Unfortunately, lack of sleep hours and poor quality of sleep have been shown to contribute to the development of obesity, along with other chronic conditions [8,9]. Interestingly, increasing sleep durations have been associated with lower rates of obesity amongst shift workers [10].

Due to these challenges, night-shift workers need appropriate lifestyle interventions aimed at alleviating the risks of poor metabolic health. Prior research has concluded that normal dietary recommendations alone are not enough to combat the negative effects of night shift [11]. Interventions that look to improve sleep, along with nutrient composition and timing, may counteract the effects of night shift [12,13,14]. We hypothesized that increasing dietary protein while advancing carbohydrate intake into the earlier active period would improve metabolic health by enhancing satiety and postprandial glycemic control, preserving lean mass, and reducing internal circadian misalignment between central and peripheral clocks [12,13]. We therefore tested whether an 8-week lifestyle intervention increasing sleep and protein intake while optimizing carbohydrate timing would decrease visceral fat and improve metabolic markers and quality of life.

## 2. Materials and Methods

### 2.1. Ethical Approval and Participant Inclusion Criteria

The study was conducted as a randomized crossover clinical trial with prior review and approval from the Auburn University Institutional Review Board and following the most recent revisions of the Declaration of Helsinki (IRB approval#: 23-496 MR 2310). The study was registered as a clinical trial prior to recruitment (ClinicalTrials.gov, registration NCT #06158204). Participants were recruited from local Auburn area healthcare facilities via email, flyers, tabling, and word of mouth. Inclusion criteria warranted that females working in healthcare must be 18 to 50 years old, have a body mass index (BMI) between 25 and 40 kg/m^2^, and have worked >6 months in a predominantly night-shift schedule. Exclusion criteria required that participants could not be pregnant or undergoing hormonal treatment for fertility, did not have any major changes in prescription medications within the last 3 months, did not have any major surgeries within the last 3 months, did not have a diagnosis of type 2 diabetes mellitus or other major endocrine diseases, did not have any dietary limitations or food allergies, and were not actively trying to lose weight or engaged in another intensive weight loss program. Participants who were eligible and agreed to participate were given an informed consent packet and were verbally informed of the study procedures. After verbal and written consent, participants completed a health history questionnaire and scheduled a time for initial testing.

### 2.2. Study Design

At the initial visit, participants arrived at the laboratory after fasting for 8 h and not working a night shift in the prior 24 h. Participants initially underwent a rapid pregnancy test and urine specific gravity (USG) to ensure appropriate hydration prior to other testing. Then, phlebotomy, height and weight measurements, and whole-body dual-energy x-ray absorptiometry (DXA) were completed. Two days of 24 h dietary recalls were completed by study staff following the multiple-pass method at this first appointment, and two more days were completed in the days following the appointment. Questionnaires to assess quality of life (QOL), mental health, and sleep were completed [15]. The same assessments were completed at the 8- and 16-week mark. Prior to leaving the initial screening visit, participants were randomly assigned to either the immediate or delayed intervention group using a computer-generated sequence created in the Sealed Envelope randomization program (Sealed Envelope Ltd., London, UK). Randomization was conducted in blocks of four and stratified by race (White and non-White) and BMI category (25–33 and 34–40).

In addition to the in-person study visits, 4 additional days of 24 h dietary recalls were completed at the 4- and 12-week mark. At the 8-week mark, participants were required to come back into the laboratory and complete the same set of tests as on the initial visit. Lastly, participants returned during week 16 for the final set of data collection.

### 2.3. Intervention

During the intervention period, participants were provided with whey protein isolate powder and grain-based snack bars to be consumed daily. At the baseline visit, a registered dietitian provided general nutrition education specific to the study intervention and developed individualized meal plans to support adherence to a eucaloric diet. Energy (~30 kcal/kg) and protein (2 g/kg) requirements were calculated from lean body mass, and plans were customized based on each participant’s habitual intake and shift schedule. Participants were advised to consume higher-carbohydrate, lower-fat meals in the evenings and higher-carbohydrate, lower-fat meals in the mornings. Whey protein isolate canisters (per serving 76 g scoop: 25 g protein, <1 g carbohydrate, 0 g fat, 110 kcal; TYM Athletic Performance, Dallas, TX, USA) and snack bars (per serving: 3 g protein, 38 g carbohydrate, 5 g fat, 200 kcal; Nature’s Bakery, Reno, NV; per serving: 6 g protein, 16 g carbohydrate, 14 g fat, 190 kcal; Member’s Mark, Bentonville, AR, USA) were provided to participants throughout the entirety of the 8-week lifestyle intervention period. The timing of the protein shakes was dependent on their work schedule, but participants were instructed to consume two whey protein shakes per day (each with 20 fl. oz. of tap water). If participants were working that night, they were instructed to consume a shake and a snack bar at approximately 11:00 p.m. and then again at approximately 5:00 a.m. On non-working days, participants were advised to consume a protein shake with their breakfast and another in the evening around 7:00 p.m. with a snack bar. Participants were encouraged to try to sleep between 10:00 a.m. and 4:00 p.m. daily, even if they were not working that evening. In addition, they received daily text messages (via the HIPAA-compliant web-based service, Emitrr) reminding them to consume the protein shakes and snack bars. Participant compliance was tracked through engagement with the messages.

### 2.4. Body Composition

To obtain baseline measurements, height and body mass of each participant were measured using a digital column scale (Seca 769; Hanover, MD, USA). Participants were instructed to lie in a supine position to complete a full-body DXA scan (Lunar Prodigy; GE Corporation, Fairfield, CT, USA) to assess lean body mass and visceral fat mass. All scans were conducted under standardized morning conditions following an overnight fast of at least 8 h and prior to any physical activity or food intake. The scans took approximately 10 min to complete and were performed by the same technicians on each visit.

### 2.5. Phlebotomy

Blood samples were aseptically obtained by a trained phlebotomist using a 23-gauge needle and K+-EDTA tube. Following collections, tubes were incubated at room temperature for approximately 30 min, and serum was separated via centrifugation at 3500× *g* for 5 min at room temperature. Serum supernatants were then placed in cryogenic 1.7 mL tubes and frozen at −80 °C until analysis. A comprehensive metabolic panel (CMP) was performed by East Alabama Medical Center to assess serum markers of metabolism as well as proteins and enzymes indicative of liver and kidney function. Assayed biomarkers included albumin, alkaline phosphatase, alanine aminotransferase (ALT), aspartate aminotransferase (AST), bilirubin (total and direct), blood glucose, blood urea nitrogen, calcium (Ca), carbon dioxide (bicarbonate), chloride (Cl), high- and low-density lipoproteins, triglycerides, creatinine and creatinine clearance, gamma-glutamyl transferase (GGT), lactate dehydrogenase, phosphate, potassium (K), sodium (Na), total serum protein, and uric acid [16].

### 2.6. Fecal Microbiome Analysis

Stool samples were collected via commode specimen collection devices and the OmniMET-Gut (ME-200 and OM-200, DNAGENOTEK) tubes. Samples were collected within 24 h of their scheduled visit, as they were stable at room temperature. However, upon receiving the sample in the lab, specimens were then stored at −80 °C until sample processing was performed.

Zymo Fecal DNA Miniprep kits (Zymo Research Inc., Irvine, CA, USA) were used to isolate fecal microbial DNA per the manufacturer’s instructions. The v4 region of the 16S rRNA gene was amplified and sequenced using Illumina Miseq (Illumina Inc., San Diego, CA, USA). The raw data was assessed for quality using FASTQC to ensure proper isolation and preparation. Low-quality data was filtered out. The Quantitative Insight into Microbial Ecology Version 2.0 (QIIME2) pipeline processed the demultiplexed FASTQ files and generated Amplicon Sequence Variants (ASVs) with DADA2 [17,18,19]. Taxonomic classification was subsequently performed using the SILVA database [20]. The ASV table was used to generate taxonomy charts. Alpha diversity metrics were computed using observed species, Shannon index, and whole-tree phylogenetic diversity. Finally, beta diversity was calculated using Bray–Curtis and unifrac clustering. QIIME2 was used to generate Principal Coordinates Analysis (PCoA) for proper analysis and visualization via grouping within color-coded plots.

### 2.7. Serum Biomarker Analysis

Plasma samples were stored frozen at −80 °C until analysis. In addition to the above-cited CMP, inflammatory markers, including interleukin 1 beta (IL-1β), TNFα, and lipopolysaccharide binding protein (LBP), were measured via ELISA assays—AFG Biosciences EK204704, EK241481, and EK710260. Blood lipids were measured in duplicate using the Cholestech LDX Analyzer (Abbott Labrotories, San Diego, CA, USA)using lipid profile cassettes [21]. 

### 2.8. Accelerometry

Participants were provided with Garmin VivoFit5 devices for the duration of the study. Study staff assisted participants with creating a Garmin account, which was linked and synced weekly using Fitabase (Small Steps Lab, LLC, San Diego, CA, USA). Total daily steps, active minutes, and moderate and vigorous activity minutes were obtained. Additionally, algorithmically determined sleep data were generated, with total sleep and time resting (lying but awake) reported. Sleep phase data was provided by the manufacturer but not included since the devices contained only accelerometers and no additional biometric sensors.

### 2.9. Subjective Data

Twenty-four-hour dietary recalls were entered into the Nutrition Data System for Research (NDSR 2022; University of Minnesota, Minneapolis, MN, USA) software and aggregated by time point. The short form health survey to assess mental and physical domains of QOL (RAND Short Form-12 item questionnaire [SF-12]) [22] was completed by study participants at each data collection point on-site.

### 2.10. Statistical Analysis

All statistical analyses were performed using SPSS v29 (IBM Corp, Armonk, NY, USA) and GraphPad Prism v10 (GraphPad Software, Boston, MA, USA), and statistical significance was established at *p* < 0.05. Continuous outcomes were analyzed using linear mixed-effects models that included group, time, and group × time (GxT) interaction as fixed effects and subject as a random effect to account for within-subject correlation. For the crossover design, treatment, period, and sequence were included as fixed effects, and carry-over effects were evaluated through treatment × sequence interactions. Greenhouse–Geisser corrections were applied when the sphericity assumption was violated, and pairwise comparisons were based on model-estimated marginal means with adjustment for multiple testing where appropriate. Given the pilot crossover design and limited sample size, all analyses were treated as exploratory. Post hoc effects were explored and adjusted for multiple comparisons using Bonferroni correction. Repeated measures ANOVA was simulated with multiple beta diversity PERMANOVA tests. Spearman correlations explored relationships among outcome variables.

## 3. Results

### 3.1. Participant Flow and Baseline Characteristics

Study accrual and retention are outlined in Figure 1. Forty-five women completed the screening survey and were contacted for follow-up. Fifteen did not meet the inclusion criteria, and seventeen were lost to follow-up. Thirteen participants completed written informed consent, and six were allocated to the immediate intervention group. Of the seven in the delayed intervention group, two withdrew consent prior to the 8-week crossover. Thus, eleven participants completed both intervention and control periods. No adverse events were reported during the study, and the two participant withdrawals were due to personal and scheduling conflicts rather than study-related issues.

Baseline characteristics of the eleven participants completing the study are displayed in Table 1. No differences were observed between randomization groups in age, weight, or BMI. During recruitment, we were made aware that potentially eligible nurses assumed ineligibility because of their Baylor Shift status, which resulted in several participants reporting an average of two shifts per week. The variation in hours worked overnight is attributed to recurring overtime for some participants.

### 3.2. Anthropometry and Quality of Life Measures

Figure 2 reports the primary outcome for this trial, which was the change in % visceral fat mass. As indicated in panel a, there were no effects of group (mean difference = −0.28 kg [95% CI −1.04 to 0.48]; *p* = 0.41), time (*p* = 0.14), or their interaction (*p* = 0.99). 

Additional body composition data are reported in Appendix A. No effects of time, group, or GxT were observed for total body mass, total fat mass, or total lean body mass. Figure 2c,d report changes in the physical and mental component scores of the RAND SF-12. A group effect was observed (*p* = 0.047), with the immediate group reporting higher physical component scores over the duration of the study (mean difference = 2.46 [95% CI 0.04 to 4.87]). Time (*p* = 0.19) and interaction (*p* = 0.81) effects were not significant. The delayed group had lower physical component scores over the duration of the study (*p* = 0.047), but there were no effects of time or a GxT interaction. Mental component scores had a large range and deviation at baseline, which diminished at 8 and 16 weeks. Nonetheless, there were no effects of the intervention on SF-12 mental component scores (mean difference = −0.41 [95% CI −4.14 to 3.32]; *p* = 0.81).

### 3.3. Metabolic and Inflammatory Markers

Secondary outcomes included serum lipids and lipoproteins, as well as several biomarkers from conventional comprehensive metabolic panels. Figure 3b shows a significant GxT interaction for LDL-C (*p* < 0.001) and a main effect of time (*p* = 0.016), suggesting that LDL-C decreased across the study period, with the delayed group showing a greater reduction during their initial intervention phase compared with the immediate group. Although total cholesterol and HDL-C did not show significant interaction or main effects (*p* > 0.05), the delayed group consistently exhibited slightly lower mean HDL-C (57.4 ± 11.3 mg/dL) compared with the immediate group (62.9 ± 11.3 mg/dL). Triglyceride concentrations were significantly higher in the delayed group (145.5 ± 23.9 mg/dL) relative to the immediate group (83.1 ± 23.9 mg/dL; 95% CI = −117.7 to −7.13 mg/dL, *p* = 0.033). Collectively, these findings suggest modest group-specific differences in lipid regulation across the crossover phases. Appendix A reports glucose, creatinine, and liver enzyme data. Alanine aminotransferase remained stable, whereas aspartate aminotransferase decreased across time, consistent with improved hepatic enzyme profiles. Appendix A shows that IL-1β, TNF-α, and LBP concentrations remained stable throughout the study, indicating no significant inflammatory response to the intervention.

### 3.4. Dietary Intake Characteristics

Figure 4 displays the average daily intake for total calories and macronutrients. Although the intervention was designed to be eucaloric, total caloric intake showed a modest, nonsignificant decline across time (*p* = 0.35), with the delayed group consuming slightly more (1516 ± 279 kcal) than the immediate group (1452 ± 279 kcal; 95% CI = −707 to 578 kcal).

Protein intake demonstrated a significant GxT interaction (*p* = 0.003), suggesting that intake patterns differed between groups over the study period. The delayed group reported a slightly higher average protein intake (78.1 ± 13.7 g) compared to the immediate group (75.6 ± 13.7 g; 95% CI = −34.2 to 29.1 g), though overall consumption remained comparable.

Fat (*p* = 0.55), carbohydrate (*p* = 0.53), and fiber (*p* = 0.77) intake showed no significant differences. Sugar intake displayed a nonsignificant trend toward higher values in the delayed group (*p* = 0.07), averaging 87.2 ± 16.5 g compared to 52.7 ± 16.5 g in the immediate group (95% CI = −72.5 to 3.7 g). Collectively, these findings indicate that total energy and macronutrient intakes remained relatively consistent throughout the intervention and control phases, with only minor fluctuations observed between groups over time.

### 3.5. Actigraphy and Sleep

Actigraphy data are presented in Figure 5. There were no significant GxT effects for daily step counts (*p* = 0.15) or active minutes (*p* = 0.19), and overall physical activity remained stable throughout the study. Significant group effects were observed for both sleep and rest duration. The immediate group exhibited longer nightly sleep (*p* = 0.002; 6.99 ± 0.51 h) compared with the delayed group (4.67 ± 0.51 h; 95% CI = 1.16 to 3.48 h). Similarly, rest hours were higher in the immediate group (*p* = 0.0006; 7.27 ± 0.46 h) relative to the delayed group (4.92 ± 0.46 h; 95% CI = 1.32 to 3.38 h). Collectively, these findings indicate that physical activity remained consistent across time, while the immediate group consistently achieved greater sleep and rest duration than the delayed group.

### 3.6. Fecal Microbiome Diversity and Composition

Alpha diversity metrics are displayed in Figure 6. No significant effects of group, time, or GxT were observed for Faith’s Phylogenetic Diversity (*p* = 0.90), Observed Species (*p* = 0.87), Shannon Entropy (*p* = 0.27), or Pielou’s Evenness (*p* = 0.28). Mean values for Faith’s PD were similar between the immediate (9.14 ± 1.22) and delayed (8.78 ± 1.22) groups (95% CI = −2.45 to 3.16), indicating stable within-sample diversity across the intervention.

Beta diversity analyses using Bray–Curtis (panel a) and Weighted UniFrac (panel b) distance matrices are shown in Figure 7. Significant group effects were detected for Bray–Curtis (*p* = 0.002) and Weighted UniFrac (*p* = 0.028), while no significant time or GxT effects were observed for any metric. Emperor plots demonstrate moderate clustering by group, suggesting subtle compositional shifts in microbial communities between the immediate and delayed groups over the study period.

### 3.7. Exploratory Correlations

Correlations among changes in outcome measures are reported in Figure 8. After correction for multiple comparisons, no significant relationships were observed, though the change in observed species correlated with the change in dietary fat (*p* = 0.049).

## 4. Discussion

This study aimed to investigate the effects of a structured 8-week dietary and sleep intervention on visceral fat mass and quality of life in night-shift workers. Our findings indicate that while some metabolic markers and dietary intake measures showed trends toward improvement, the overall impact on body composition was inconsistent. These results highlight the complexity of mitigating the adverse health effects of night-shift work through lifestyle interventions alone.

### 4.1. Metabolic and Inflammatory Responses

Previous studies have established that shift workers experience circadian misalignment, which contributes to metabolic dysfunction, increased adiposity, and a higher risk of chronic conditions such as obesity, type 2 diabetes, and cardiovascular disease [23]. Despite an observed increase in protein intake among participants (*p* < 0.001), no significant reductions in visceral fat mass were detected (*p* = 0.963). This finding is consistent with other studies that suggest that short-term dietary modifications alone may not be sufficient to counteract the metabolic disturbances caused by circadian misalignment [24]. Circadian misalignment alters peripheral clock gene expression, particularly in hepatic and adipose tissues, leading to impaired glucose tolerance, dyslipidemia, and increased visceral adiposity [6,7]. Observational studies confirm that night-shift workers consume more energy at biologically inappropriate times and exhibit higher inflammatory dietary indices and poorer sleep compared to day workers [1,4]. These physiological and behavioral disruptions contribute to the elevated prevalence of obesity and metabolic disease observed in this population.

### 4.2. Dietary Intake and Macronutrient Distribution

The intervention successfully increased protein intake in the immediate intervention group, reaching an average of 101.3 g/day by week 4, but no significance was observed across groups (*p* = 0.858). High-protein diets have been shown to promote satiety and lean mass retention in individuals undergoing caloric restriction [25]. However, the reduction in carbohydrate intake was not as pronounced as anticipated (*p* = 0.686), which may have limited the metabolic benefits of macronutrient redistribution. Studies have suggested that carbohydrate restriction at night can improve metabolic flexibility and glycemic control in shift workers, but compliance with such dietary recommendations remains a challenge in real-world settings [26].

Previous work has demonstrated that the timing of food intake itself plays a critical role in metabolic regulation. In a controlled laboratory study, Arble et al. (2009) showed that mice fed during their usual rest phase (the light cycle) gained significantly more weight than those fed the same diet during their active phase, despite equivalent caloric intake, indicating that circadian timing of meals directly influences energy storage and adiposity [24]. Similarly, Spaeth et al. (2013) experimentally restricted sleep in healthy adults and found that short-sleep conditions led to greater caloric intake, particularly from late-night snacking, and progressive weight gain over a one-week period [27]. These studies highlight that both feeding time and sleep duration independently modulate energy balance, which may help explain the limited body composition changes observed in our short-term intervention. Further, it is of note that the dietary data was based on 24 h dietary recalls performed with the participants. That is a significant limitation to the findings, as there might be errors in recollection.

### 4.3. Impact on Body Composition and Physical Health

Despite the intervention of dietary modifications, no significant changes in total body mass (*p* = 0.963), fat mass (*p* = 0.362), or lean body mass (*p* = 0.790) were observed over the 16-week intervention period. This aligns with research indicating that dietary interventions alone may not be sufficient to produce substantial changes in body composition without concurrent structured physical activity programs [28]. Evidence from a systematic review and meta-analysis by Johns et al. (2014) further supports this conclusion, showing that combined diet and exercise interventions produce significantly greater reductions in body weight (mean difference ≈ −1.7 kg), waist circumference, and improvements in insulin sensitivity compared with diet- or exercise-only interventions [29]. Similarly, Nepper et al. (2020) demonstrated that a behavioral lifestyle program incorporating both nutrition education and physical activity counseling led to significant improvements in body composition and metabolic health outcomes in overweight and obese healthcare workers, highlighting the real-world efficacy of combined interventions [30]. Collectively, these findings underscore that integrating structured exercise with dietary modification amplifies metabolic adaptations through enhanced energy expenditure, muscle insulin sensitivity, and fat oxidation, factors that may be necessary to elicit meaningful body composition changes in shift-working populations.

### 4.4. Quality of Life and Mental Health

While SF-12 scores did not show significant improvements in mental health components (*p* = 0.792), there was a notable difference between groups (*p* = 0.047) in the physical component score (PCS). This suggests that participants who adhered to the intervention may have perceived some benefits in physical well-being, even if objective measures did not reflect significant changes. Sleep quality improvements have been associated with enhanced overall well-being and cognitive function, emphasizing the need for future studies to explore additional sleep hygiene strategies alongside dietary interventions [27]. Including sleep hygiene and exercise changes in conjunction with dietary modifications may bolster changes to PCS and MCS measures for participants.

### 4.5. Gut Microbiome Outcomes

Gut microbial diversity was assessed using alpha and beta diversity metrics. Alpha diversity remained stable with no significant differences over time or between groups, suggesting resilience of microbial richness to short-term interventions. This finding is consistent with prior work showing limited changes in alpha diversity under acute circadian disruption, such as short-term night-shift exposure or partial sleep deprivation, where metabolic impairments occurred without major microbial diversity shifts [31,32].

In contrast, beta diversity analyses revealed significant between-group differences (Bray–Curtis *p* = 0.002; Weighted UniFrac *p* = 0.028), indicating that the immediate and delayed intervention groups differed in microbial community composition. However, these differences did not change over time, suggesting stable but distinct community structures between groups. Similar observations have been reported in circadian misalignment models, where microbial rhythmicity was disrupted alongside metabolic dysregulation [33]. Such shifts may reflect the bidirectional relationship between circadian rhythms and the gut microbiome, with certain taxa exhibiting diurnal oscillations that influence host metabolic processes [34,35]. Experimental disruption of these microbial rhythms has been linked to impaired glucose tolerance and altered energy balance [33,36]. These results indicate that while overall microbial richness remained stable, group-specific compositional shifts were present and may reflect underlying circadian disruption.

### 4.6. Limitations and Future Directions

Several limitations should be acknowledged. The small sample size (n = 11) limits the generalizability of our findings due to a small sample size. Additionally, participant adherence to dietary recommendations was self-reported, introducing potential reporting biases. The study did not include chronobiological biomarkers (e.g., melatonin or cortisol), which limits the interpretation of circadian alignment; future studies should address this limitation. Future studies should also explore the impact of longer intervention durations, larger sample sizes, and the inclusion of structured exercise programs to maximize health benefits in night-shift workers. Additionally, including proper sleep hygiene to ideally improve circadian misalignment would be worth further evaluation in future studies.

## 5. Conclusions

This study contributes to the growing body of literature on lifestyle interventions for night-shift workers by highlighting the challenges of modifying dietary intake and sleep patterns to improve metabolic health. While some dietary improvements were achieved, significant changes in visceral fat mass and inflammatory markers were not observed, underscoring the need for multi-faceted intervention approaches. Future research should focus on integrating physical activity and personalized sleep interventions to optimize metabolic outcomes in this vulnerable population.

## Figures and Tables

**Figure 1 nutrients-17-03342-f001:**
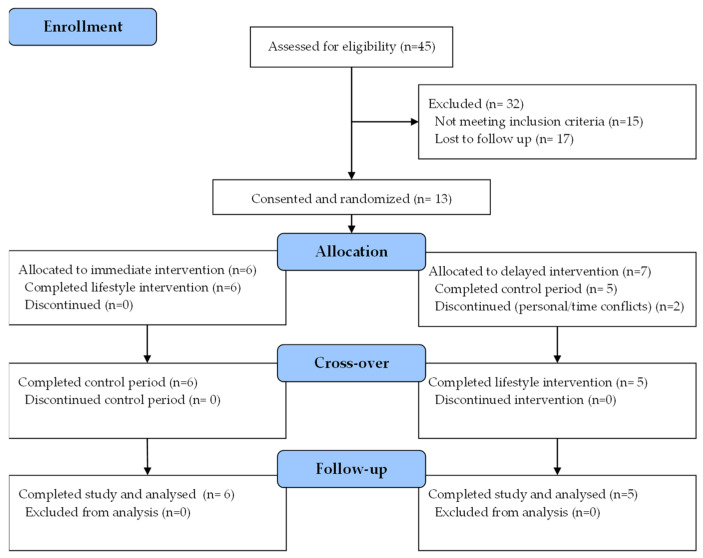
CONSORT diagram.

**Figure 2 nutrients-17-03342-f002:**
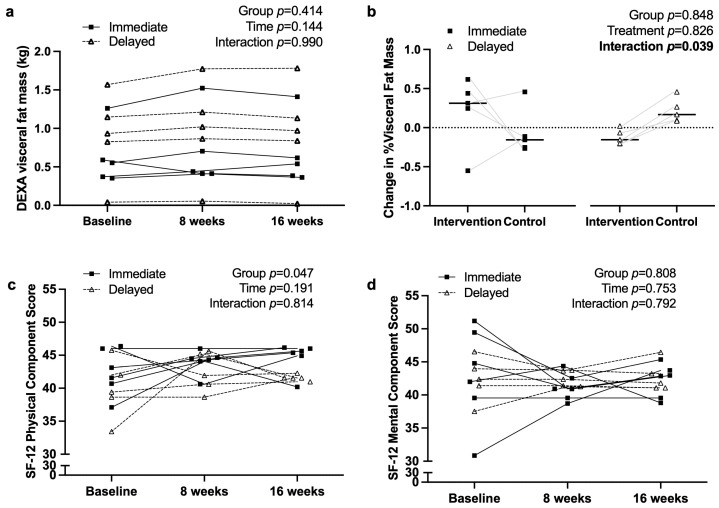
Individual changes in visceral fat mass measured by DEXA and reported (**a**) actual visceral fat mass by time point and (**b**) % change in visceral fat mass during each period of the study. Changes in quality of life as measured by SF-12: (**c**) Physical Health Component Score and (**d**) Mental Health Component Score.

**Figure 3 nutrients-17-03342-f003:**
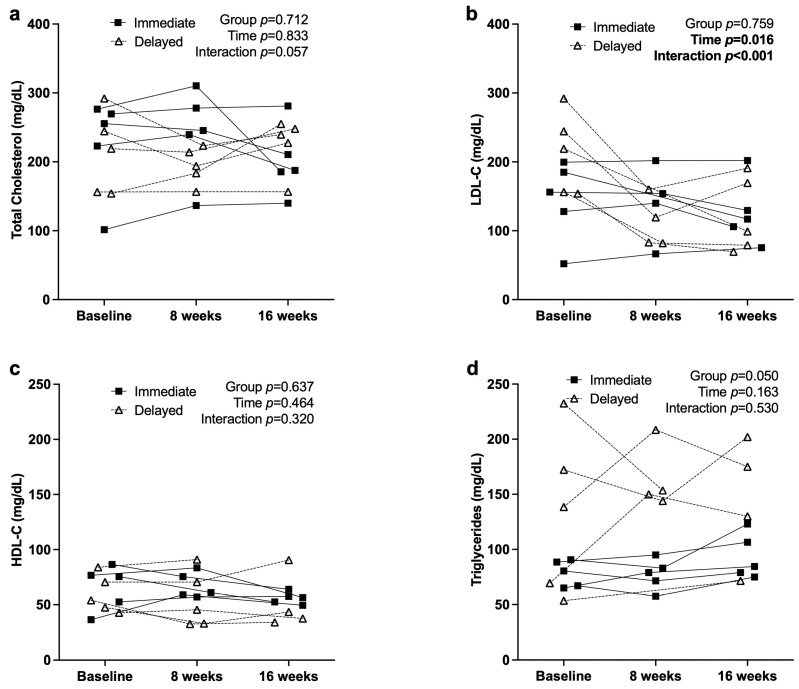
Individual changes in serum biomarkers: (**a**) total cholesterol, (**b**) LDL-C, (**c**) HDL-C, and (**d**) triglycerides at baseline, 8 weeks, and 16 weeks.

**Figure 4 nutrients-17-03342-f004:**
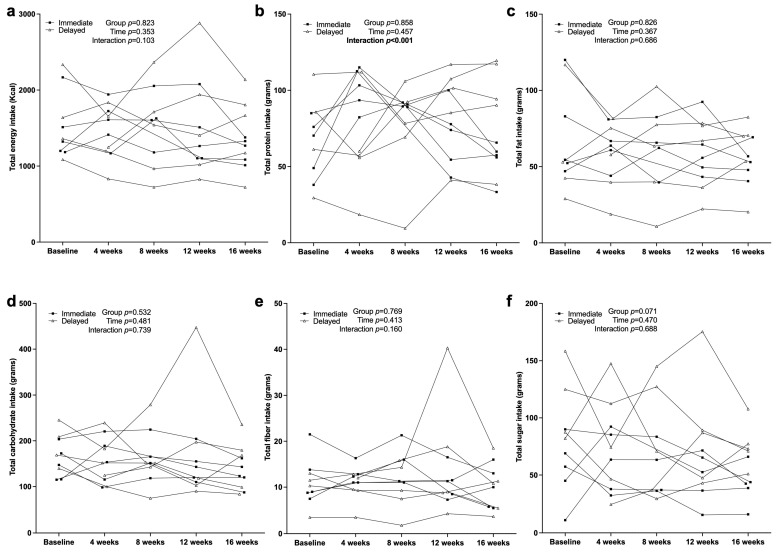
Individual nutrient intake trajectories measured by 24 h dietary recall every 4 weeks: (**a**) total energy intake, (**b**) total protein intake, (**c**) total fat intake, (**d**) total carbohydrate intake, (**e**) total fiber intake, and (**f**) total sugar intake.

**Figure 5 nutrients-17-03342-f005:**
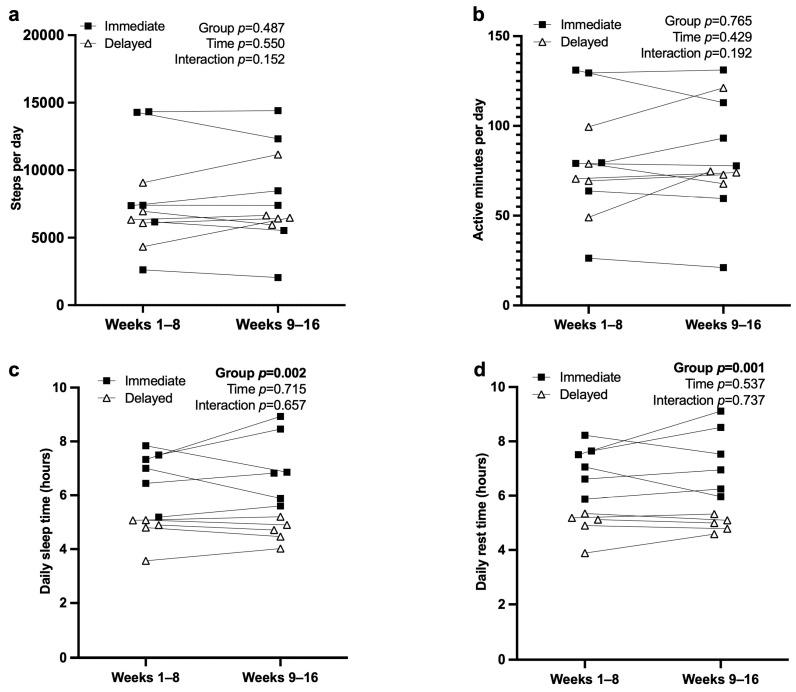
Individual actigraphy measures in immediate and delayed groups: (**a**) steps, (**b**) active minutes, (**c**) sleep, and (**d**) rest across study phases (weeks 1–8 and 9–16).

**Figure 6 nutrients-17-03342-f006:**
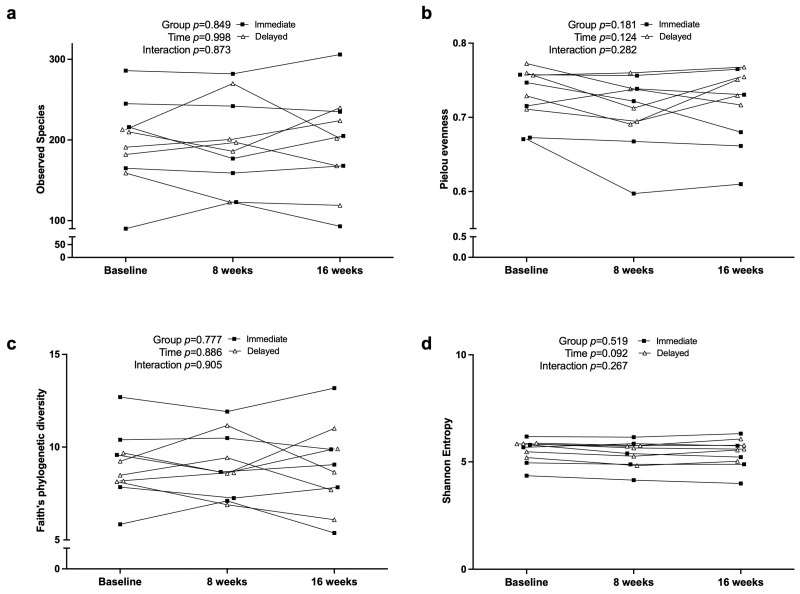
Alpha diversity metrics for individual study participants across baseline (T00), crossover (T08), and follow-up (T16): (**a**) Observed Species, (**b**) Pielou Evenness, (**c**) Faith’s Phylogenetic Diversity, and (**d**) Shannon Entropy.

**Figure 7 nutrients-17-03342-f007:**
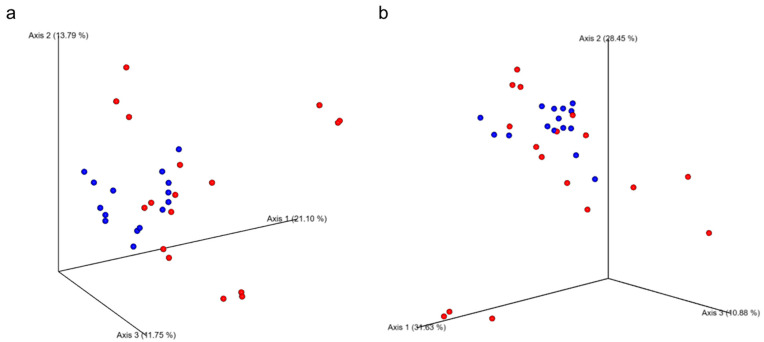
Emperor plots of (**a**) Bray–Curtis (*p* = 0.002) and (**b**) Weighted Unifrac (*p* = 0.028) principal coordinates comparing immediate (red) and delayed (blue) groups.

**Figure 8 nutrients-17-03342-f008:**
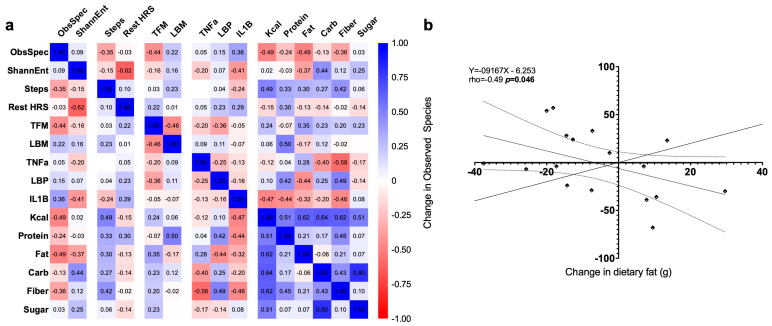
(**a**) Heatmap showing correlations among changes in study outcome variables; (**b**) Relationship between change in dietary fat intake and change in gut microbial richness (Observed Species). Each point represents an individual study period (i.e., one participant per intervention condition). The solid line represents the best-fit linear regression, with dotted lines indicating 95% confidence intervals.

**Table 1 nutrients-17-03342-t001:** Participant Characteristics.

Measure	All (n = 11)	Immediate (n = 6)	Delayed (n = 5)	*p* Value
Age (years)	32.6 (8.5)	31.2 (10.0)	34.4 (7.0)	0.279
Weekly Shifts	3.3 (1)	3.7 (1.2)	2.8 (0.4)	0.083
Hours Worked Overnight	11.6 (1.2)	11.3 (1.6)	12 (0.0)	0.195
Dual X-ray Absorptiometry				
Total Body Mass (kg)	80.09 (12.12)	78.51 (11.69)	81.67 (13.7)	1.000
Lean Body Mass (kg)	44.65 (3.63)	44.9 (4.27)	44.4 (3.36)	0.996
Total Fat Mass (kg)	32.78 (9.68)	32.15 (10.11)	33.41 (10.38)	1.000
Accelerometry				
Daily Steps	7725 (3659)	8698 (4684)	6558 (1709)	0.347
Daily Rest Time (hours)	6.12 (1.37)	7.15 (0.83)	4.89 (0.58)	<0.001
Average dietary intake				
Energy (Kcal)	1534 (442)	1477 (408)	1284 (855)	1.000
Protein (g)	67.3 (25.7)	63.7 (19.5)	57.4 (43.9)	1.000
Fat (g)	66.5 (32.7)	71.4 (30.6)	48.3 (43.2)	0.996
Carbohydrate (g)	168.5 (44.3)	150.8 (38)	152.5 (94.2)	0.703
Fiber (g)	11 (5)	12.1 (5.8)	7.7 (5.6)	1.000
Sugar (g)	80.6 (43.2)	54.5 (29.4)	90.6 (59.3)	0.194

Baseline data of 11 participants completing the study. Values are mean (standard deviation).

## Data Availability

The datasets generated and analyzed during the current study are not publicly available due to ethical and confidentiality restrictions related to participant privacy and the small cohort size. De-identified data that support the conclusions of this study are available from the corresponding author upon reasonable request.

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
