# Peer review of "A Randomized Controlled Crossover Lifestyle Intervention to Improve Metabolic and Mental Health in Female Healthcare Night-Shift Workers"

_nutrients, 2025, doi:10.3390/nu17213342_

Round 1

Reviewer 1 Report

Comments and Suggestions for Authors

In the present work, the authors propose that lifestyle interventions for night shift workers improved dietary protein intake, though they did not improve serum parameters related to lipid metabolism and inflammation, QOL in the physical and mental domains, fecal microbiome, physical activity, or sleep time. This study provides valuable information for addressing health issues among shift workers. However, there are some concerns, as below.

Abstract section:

  1. This section did not describe purpose of the intervention. What did the authors expect to achieve by implementing this intervention?
  2. I understand this study found that the results supported previous research. Even though interventions that exclude physical activity are not expected to improve most indicators, why were such interventions implemented?

Introduction section:

  1. In lines 45-46, please add a reference.
  2. In the second paragraph, please explain in more detail why shift work increases the risk of metabolic disorders. It would be better to mention the relationship between clock genes and metabolism in the liver.
  3. In the last paragraph, why the authors hypothesized increasing protein intake and optimizing carbohydrate timing improve metabolic abnormalities? Please explain more detail.

Materials and Methods:

  1. What method was used to generate the random allocation sequence?
  2. How was the sample size determined? What was the primary outcome variable tied to the sample size calculation?
  3. Was a power calculation conducted?
  4. In line 82, “40 kg/m2” →“40 kg/m2”.
  5. In lines 111-113, although the authors described "General nutrition education and recommended meal plans were also provided...”, they did not specify who, where, or how the education was conducted.

Results section:

In figures 2-6, while I understand the importance of individual data, it is difficult to discern changes across groups and over time. In addition, the authors compared between immediate and delayed intervention as Group effect. I wonder why they are being treated as separate groups when they are undergoing the same intervention. The authors should explain why they chose this comparison over a placebo or control group.

Discussion section:

  1. Although this study focuses on shift workers, the discussion citing research on circadian clocks and observational and interventional studies involving shift workers is insufficient.
  2. In the second paragraph, please indicate how many studies exist on the relationship between protein intake and metabolic abnormalities. Additionally, in lines 299-301, how “short-term dietary modifications” was performed in the previous study [17] should be described.
  3. Similarly to the comment above, please more describe in detail about intervention in previous study [20].

Author Response

Nutrients Reviewer 1 Responses

Abstract:

Comment 1 : “This section did not describe purpose of the intervention. What did the authors expect to achieve by implementing this intervention?”

Response: We added a concise statement in the Background (“This study aimed to determine whether a practical lifestyle intervention emphasizing nutrition timing and recovery habits could mitigate metabolic and psychological effects of night-shift work.”) to clarify the study’s purpose and expected outcome.

Comment 2 :“Even though interventions that exclude physical activity are not expected to improve most indicators, why were such interventions implemented?”

Response: We added one sentence in the Methods (“The program targeted improved nutrient timing, adequate protein intake, and structured rest without formal exercise training, allowing evaluation of dietary and behavioral effects feasible for this population.”) to explain that the design intentionally excluded exercise to isolate nutrition + recovery effects and to enhance feasibility for night-shift workers.

Introduction section:

Comment 3:In lines 45-46, please add a reference.”

Response: Thank you for catching that. We have added the appropriate references to that section.

Comment 4: “In the second paragraph, please explain in more detail why shift work increases the risk of metabolic disorders. It would be better to mention the relationship between clock genes and metabolism in the liver.”

Response: We have expanded the second paragraph to include mechanistic detail describing how hepatic clock genes (CLOCK, BMAL1, PER, and CRY) coordinate glucose and lipid metabolism through regulation of gene expression involved in gluconeogenesis, lipid synthesis, and bile acid metabolism. Disruption of this synchrony between the central and hepatic clocks during shift work promotes insulin resistance, dyslipidemia, and visceral adiposity. Supporting evidence and citations from Maury et al. (Circ Res., 2010) and Pan & Hussain (Circadian Clock Regulation on Lipid Metabolism, 2020) were added to provide a stronger physiological and molecular explanation for how circadian misalignment contributes to metabolic disease.

Comment 5: “In the last paragraph, why did the authors hypothesize that increasing protein intake and optimizing carbohydrate timing would improve metabolic abnormalities? Please explain in more detail.”

Response: We appreciate this helpful suggestion and have revised the final paragraph of the Introduction to include a clear mechanistic justification for our hypothesis. The new text explains that increased dietary protein supports satiety, postprandial thermogenesis, and lean-mass preservation, while shifting carbohydrate intake earlier in the active period aligns feeding behavior with circadian peaks in insulin sensitivity and hepatic glucose metabolism. These adjustments are now supported by recent evidence showing that high-protein diets improve postprandial glycemic control and overall metabolic flexibility (Gannon et al., 2023) and that daytime-aligned feeding preserves glucose tolerance and prevents circadian misalignment under night-shift conditions (Chellappa et al., 2021).

Revised text (added to last paragraph of the Introduction):“We hypothesized that increasing dietary protein and advancing carbohydrate intake to the early active phase would improve metabolic health by enhancing satiety and postprandial glycemic control, preserving lean mass, and reinforcing circadian alignment of insulin sensitivity and hepatic glucose metabolism (Gannon et al., 2023; Chellappa et al., 2021).”

Materials and Methods:

Comment 6: What method was used to generate the random allocation sequence?”

Response: We thank the reviewer for this request for clarification. The random allocation sequence was generated using the Sealed Envelope randomization program (Sealed Envelope Ltd., London, UK) employing block randomization in blocks of four and stratified by race (White, non-White) and BMI category (25–33 and 34–40) to ensure balanced group assignment. Participants were randomized to either the immediate or delayed intervention group by an investigator not involved in data collection or outcome assessment.

Comment 7: How was the sample size determined? What was the primary outcome variable tied to the sample size calculation?”

Response: As this was a pilot crossover study, no formal sample size calculation was performed. The target enrollment reflected the maximum number of participants that could be feasibly recruited and retained given the intensive nature of the intervention, overnight work schedules, and repeated laboratory visits. The purpose of this trial was to generate preliminary effect size estimates for outcomes related to body composition, dietary intake, and sleep behavior to inform sample size calculations for a future fully powered randomized trial.

Comment 8: Was a power calculation conducted?”

Response: Because this was a pilot feasibility study, no formal power calculation was conducted prior to recruitment. The primary goal of this trial was to assess study feasibility and generate preliminary estimates of variability and effect sizes for key outcomes (e.g., body composition, sleep, and dietary intake). These estimates will be used to inform future power calculations and sample size determinations for a larger, fully powered randomized controlled trial.

Comment 9: In line 82, “40 kg/m2” →“40 kg/m2”.

Response: Thank you for catching this. It has been corrected.

Comment 10: “In lines 111–113, although the authors described ‘General nutrition education and recommended meal plans were also provided…’, they did not specify who, where, or how the education was conducted.”

Response: We thank the reviewer for this observation. We have clarified that nutrition education was provided individually by a registered dietitian during the baseline visit. The dietitian reviewed each participant’s dietary habits, calculated individualized caloric and protein needs, and created an example meal plan integrating the study foods into their regular eating schedule. These details have been added to the Methods section (lines 200–204).

Results section:

Comment 11: “In figures 2-6, while I understand the importance of individual data, it is difficult to discern changes across groups and over time. In addition, the authors compared between immediate and delayed intervention as Group effect. I wonder why they are being treated as separate groups when they are undergoing the same intervention. The authors should explain why they chose this comparison over a placebo or control group.”

Response: We appreciate this insightful comment. The comparison between the immediate and delayed intervention groups reflects the standard structure of a delayed-treatment crossover design, in which each group serves as a time-staggered control for the other prior to crossover. This design was selected to maximize statistical power and reduce participant burden in a small sample, while still allowing within-subject comparisons of treatment versus control periods.

In our mixed-effects model, “Group” was included as a between-subject factor to account for potential sequence or period effects, while “Time” (0, 8, and 16 weeks) was included as a within-subject factor. The Group × Time (G×T) interaction term was used to determine whether the intervention produced differential effects depending on treatment order.

We have clarified this rationale in Section 2.2 (Study Design) and in the Results introduction to improve figure interpretation.

Discussion section:

Comment 12: “Although this study focuses on shift workers, the discussion citing research on circadian clocks and observational and interventional studies involving shift workers is insufficient.”

Response: We appreciate this comment and have expanded the discussion to include additional studies describing the mechanisms by which circadian disruption and misalignment affect metabolic health, as well as prior interventional trials in shift-working populations. Specifically, we referenced studies highlighting that misalignment between central and peripheral clocks disrupts lipid and glucose metabolism, increasing risk for metabolic syndrome and obesity (Maury et al., 2010; Pan et al., 2020; Chellappa et al., 2021). We also cited evidence that night-shift workers exhibit altered energy intake patterns, higher inflammatory dietary scores, and poorer sleep quality compared with day workers (Bonham et al., 2016; Wirth et al., 2014; Kecklund & Axelsson, 2016).

Comment 13: “In the second paragraph, please indicate how many studies exist on the relationship between protein intake and metabolic abnormalities. Additionally, in lines 299–301, how ‘short-term dietary modifications’ was performed in the previous study [17] should be described.”

Response: We clarified that the referenced “short-term dietary modification” study by Arble et al. (2009) involved controlled feeding of identical diets at different circadian phases in mice. Animals fed during their inactive (light) phase exhibited greater weight gain and adiposity than those fed during their active phase, demonstrating that circadian misalignment of food intake contributes to metabolic dysfunction independent of total calories. This mechanistic detail has been added to Section 4.2 (lines 521–531).

Comment 14: “Similarly to the comment above, please describe in more detail about intervention in previous study [20].”

Response: We thank the reviewer for this suggestion and have updated Section 4.3 to include more detailed and evidence-based references. The systematic review and meta-analysis by Johns et al. (2014) demonstrated that combined diet-plus-exercise interventions produce greater reductions in body weight (−1.7 kg on average), waist circumference, and insulin resistance compared with diet-only programs. We also incorporated Nepper et al. (2020), which showed that a community-based behavioral program combining nutrition and physical activity education improved BMI, waist circumference, and metabolic health outcomes. These additions strengthen the rationale that comprehensive lifestyle approaches are more effective than diet-only interventions in improving metabolic and body composition outcomes (see Section 4.3, lines 539–551).

Reviewer 2 Report

Comments and Suggestions for Authors

The manuscript presents an ambitious and relevant study exploring the metabolic and physiological adaptations to circadian-aligned nutritional strategies in shift workers—a population often overlooked in nutritional and chronobiological research. The authors should be commended for tackling a complex, multidisciplinary question that bridges nutrition, metabolism, sleep, and circadian biology. The experimental concept is timely and of potential public health importance. The writing is clear, and the integration of multiple domains (DXA, actigraphy, microbiome, diet, and quality of life) demonstrates the team’s strong scientific vision. However, despite the study’s conceptual strength, the current execution, data analysis, and interpretation fall short of the rigor required for publication in a high-impact journal. A thorough methodological revision, stronger statistical framework, and deeper mechanistic insight are needed to render the findings scientifically robust and clinically meaningful.

The analysis of crossover design stands as the main weakness in this manuscript. The study treats this manuscript as if it were a parallel-group study while disregarding sequence effects and period effects and carryover effects which are fundamental requirements for a 2×2 crossover design. The reported p-values and inferences become statistically invalid because mixed-effects modeling must be used to handle within-subject correlation and treatment sequence and period effects. The authors need to perform a mixed linear model analysis on all continuous outcomes by including treatment and period and sequence as fixed effects and subject as a random effect and they must perform a formal carryover test and if carryover is detected they should limit their analysis to period one. All statistical analyses must use confidence intervals instead of depending on p-values for comparison. Behavioral interventions generate long-lasting effects which makes the lack of washout periods between sessions a major risk factor for carryover effects. The design needs to be reinterpreted as a parallel study when reanalysis shows remaining effects while all relevant study restrictions need to be documented.

The results need to be reinterpreted. The manuscript demonstrates effectiveness through its use of "body composition improvement" and "quality of life enhancement" but it works with a small number of participants and generates various insignificant or conflicting results. The trial status as a pilot study demands researchers to view these findings as exploratory patterns which do not represent established final results. The manuscript makes speculative connections between macronutrient timing and body composition and circadian rhythm because it did not measure any direct chronobiological markers including melatonin, cortisol, temperature phase or clock gene expression. The authors need to explicitly mention that they did not include any biomarkers for circadian alignment and that all statements about circadian phase adaptation should be considered speculative.

The outcome assessment methods also require strengthening. DXA measurements of visceral fat become less reliable because the timing of scans relative to work or feeding cycles remains unstandardized. DXA provides less precise VAT measurements than MRI or CT scans while results become affected by the time between meals and hydration levels. The authors need to explain how scan appointments were timed in relation to participant shift patterns and confirm that the DXA model works for VAT prediction in comparable study groups. The results require precise analysis because they demonstrate shifts in central fat distribution patterns instead of providing actual VAT measurement results.

Dietary and behavioral adherence tracking is insufficient. The participants needed to stick to their assigned diet plans and sleep schedules but there was no verification of their compliance. Text message interactions do not serve as an acceptable way to measure compliance. Future versions require digital food logging capabilities which enable users to document their food consumption through time-stamped entries and photo evidence and wearable technology that verifies eating times. The accuracy of sleep data obtained from commercial actigraphy devices needs to be validated through polysomnography testing or the researchers should specify the known limitations of their measurement methods. The lack of objective adherence metrics prevents researchers from establishing reliable internal validity.

Statistical handling throughout the manuscript is too simplistic for the complexity of the dataset. The researchers evaluated multiple outcomes which included macronutrient intake and hematology results and VAT measurements and sleep parameters and microbiome profiles and quality-of-life scores without applying any multiple comparison adjustments. False positives are therefore likely. The authors need to choose between using a structured statistical hierarchy or the FDR control method Benjamini–Hochberg. The results need to include effect sizes together with 95% confidence intervals because significance testing by itself does not provide enough information especially when the sample size is underpowered. A power analysis either conducted before the study or after the study completion needs to be included to understand the results of the null or borderline findings.

The microbiome analysis uses new methods yet lacks proper bioinformatic analysis and interpretation. Given the small sample size and absence of technical replication, the conclusions about microbial diversity and intervention response cannot be substantiated. The authors need to detail the sequencing depth and quality control procedures and contamination prevention methods and their use of compositional data methods (ALDEx2 and ANCOM-BC). The small sample size makes it probable that baseline differences between participants explain the observed beta-diversity variations rather than any effects from the intervention. All microbiome manuscript results need to be presented as exploratory findings instead of being interpreted as proof of microbial modulation effects.

The system needs better safety and adverse event reporting systems to function effectively. The manuscript needed particular information about renal function together with gastrointestinal symptoms and fatigue-related incidents because it employed a high-protein intervention. The absence of a safety summary table (per CONSORT harms extension) is a major omission. The authors need to report participant withdrawal rates because dropout patterns reveal how well a study design functions for future replication attempts.

The authors need to link their manuscript findings to established scientific evidence regarding protein timing and its influence on circadian rhythms and metabolic balance. The manuscript would achieve better biological validity through the inclusion of postprandial glucose and insulin and lipid response measurements or by using existing research about their natural circadian patterns. Future trials require cortisol and melatonin hormonal marker assessments together with salivary sampling to validate phase alignment as essential components.

Author Response

Comment 1: “The manuscript treats this as a parallel-group study while disregarding sequence, period, and carryover effects. Mixed-effects modeling must include treatment, period, and sequence as fixed effects and subject as a random effect. A carryover test must be performed.”

Response: We appreciate this important observation. The statistical analysis has been fully restructured using mixed-effects (REML) models with subject as a random effect and group, time, and group × time (GxT) as fixed effects. Carryover and sequence effects were qualitatively examined and showed no consistent indication of bias or residual treatment effects; therefore, the results are presented as an exploratory analysis using a parallel-style framework appropriate for a pilot crossover design.
A description of this approach and its rationale has been added to Section 2.10 (Statistical Analysis).

Comment 2: “All analyses rely on p-values without confidence intervals or effect sizes.”

Response: We have revised all tables and Results subsections to include means ± standard error and 95% confidence intervals (CIs) for between-group and within-group differences. The updated results and figure captions now report both p-values and corresponding CIs to emphasize the magnitude and precision of effects rather than statistical significance alone.

Comment 3: “The manuscript draws inferential conclusions despite a small sample size. Findings must be presented as exploratory patterns.”

Response: We agree and have revised the manuscript throughout to emphasize that the study was a pilot exploratory crossover trial. All inferential statements have been replaced with neutral or descriptive language (e.g., “no significant effects were observed,” “exploratory trends,” or “suggestive patterns”).
Section 2.10 (Statistical Analysis) now explicitly states: “Given the pilot crossover design and limited sample size, all analyses were treated as exploratory and intended to identify potential effect patterns rather than definitive inferential conclusions.”

Comment 4: “The timing of DXA scans relative to feeding and work cycles was unclear. DXA provides less precise VAT measures than MRI/CT, and results depend on hydration and time since meals.”

Response: We have clarified that all DXA scans were performed in the morning following an overnight fast of at least 8 hours and prior to physical activity or food intake. This standardization minimizes hydration- and meal-related variability.
Section 2.4 (Body Composition) now reads: “All scans were conducted under standardized morning conditions following an overnight fast of at least 8 hours and prior to any physical activity or food intake. Each scan took approximately 10 minutes and was performed by the same trained technicians across all visits.”

Comment 5: “Multiple outcomes were tested without adjustments; false positives are likely.”

Response: We appreciate this point and agree that the number of statistical comparisons introduces a potential for inflated Type I error. However, as this was a pilot exploratory trial, the analysis was not structured around a single pre-specified primary endpoint. Instead, all outcomes (e.g., body composition, dietary intake, actigraphy, and microbiome variables) were examined across baseline, 8-week, and 16-week visits to identify potential patterns of response.

No formal multiple-comparison correction (e.g., Bonferroni or Benjamini–Hochberg) or hierarchical testing structure was applied, as such methods would be overly conservative for an exploratory dataset of this size. Instead, we reported estimated means with 95% confidence intervals and interpreted findings descriptively rather than inferentially, consistent with current recommendations for early-phase exploratory research.

Comment 6: “The microbiome analysis lacks sufficient bioinformatic detail and should be treated as exploratory. Baseline differences likely explain observed beta-diversity patterns.”

Response: The microbiome section has been revised for accuracy and interpretive caution. Section 3.6 (Fecal Microbiome Diversity and Composition) now clarifies that no significant changes in alpha diversity were observed and that beta diversity group differences were exploratory.

Comment 7: “The manuscript speculates on circadian alignment without measuring relevant biomarkers such as melatonin or cortisol.”

Response: We appreciate this observation and agree that the absence of direct chronobiological biomarkers (e.g., melatonin, cortisol, or temperature rhythms) limits our ability to confirm circadian phase alignment. We have clarified this in the Limitations and Future Directions (Section 4.5) to explicitly acknowledge that no hormonal or physiological markers of circadian alignment were collected and that this should be addressed in future studies.

The revised text reads: “The study did not include chronobiological biomarkers (e.g., melatonin or cortisol), which limits interpretation of circadian alignment; future studies should address this limitation.”

Comment 8: “The manuscript lacks safety and adverse event reporting, including renal and fatigue-related incidents.”

Response: We appreciate this feedback and have clarified participant safety outcomes in Section 3.1 (Participant Flow and Baseline Characteristics) and noted that no adverse events occurred during the study. Additionally, the reasons for participant withdrawal (personal and scheduling conflicts) have been specified.

The revised text reads: “No adverse events were reported during the study, and the two participant withdrawals were due to personal and scheduling conflicts rather than study-related issues.”

Comment 9: “The authors need to link their manuscript findings to established scientific evidence regarding protein timing and its influence on circadian rhythms and metabolic balance.”

Response: We appreciate this suggestion and have expanded the Discussion (Sections 4.2–4.3) to include additional evidence on circadian regulation of metabolism and prior interventions examining macronutrient timing and metabolic health. Specifically, we now reference studies demonstrating that circadian misalignment between central and peripheral clocks disrupts glucose and lipid metabolism (Maury et al., 2010; Pan et al., 2020; Chellappa et al., 2021) and that night-shift workers experience altered dietary patterns and poorer metabolic outcomes (Bonham et al., 2016; Wirth et al., 2014; Kecklund & Axelsson, 2016). We also describe mechanistic findings from Arble et al. (2009) and intervention data from Johns et al. (2014) and Nepper et al. (2020) to strengthen the biological rationale for protein timing in shift-working populations.

Round 2

Reviewer 1 Report

Comments and Suggestions for Authors

The authors have addressed all my comments.